# *Zanthoxylum piperitum* Benn. Attenuates Monosodium Urate-Induced Gouty Arthritis: A Network Pharmacology Investigation of Its Anti-Inflammatory Mechanisms

**DOI:** 10.3390/ph18010029

**Published:** 2024-12-29

**Authors:** Sung Wook Kim, Soo Hyun Jeong, Jong Uk Kim, Mi Hye Kim, Wonwoong Lee, Cheol-Jung Lee, Tae Han Yook, Gabsik Yang

**Affiliations:** 1College of Korea Medicine, Woosuk University, Jeonju-si 54986, Republic of Korea; sheep.sw91@gmail.com (S.W.K.); sammytega1@naver.com (S.H.J.); ju1110@hanmail.net (J.U.K.); kimmh526@woosuk.ac.kr (M.H.K.); 2Department of Pharmacy, College of Pharmacy, Woosuk University, Wanju 55338, Republic of Korea; wwlee@woosuk.ac.kr; 3Research Center for Materials Analysis, Korea Basic Science Institute, Daejeon 34133, Republic of Korea; veritas0613@kbsi.re.kr

**Keywords:** *Zanthoxylum piperitum*, NLRP3 inflammasome, monosodium urate crystals, gouty arthritis, network pharmacology

## Abstract

**Background:** Monosodium urate crystal accumulation in the joints is the cause of gout, an inflammatory arthritis that is initiated by elevated serum uric acid levels. It is the most prevalent form of inflammatory arthritis, affecting millions worldwide, and requires effective treatments. The necessity for alternatives with fewer side effects is underscored by the frequent adverse effects of conventional therapies, such as urate-lowering drugs. IL-1β is a potential therapeutic target due to its significant role in the inflammatory response induced by MSU. *Zanthoxylum piperitum* Benn. (ZP), a shrub that possesses antibacterial, antioxidant, and anti-inflammatory properties, has demonstrated potential in the treatment of inflammatory conditions. **Methods:** For anti-inflammatory properties of ZP, Raw264.7 cell stimulated LPS were treated ZP and using RNA-seq with Bone marrow derived macrophage, we observed to change inflammatory gene. Pharmacological networks were conducted to select target gene associated with ZP. For in vivo, mice were injected MSU in footpad for induce gouty arthritis model. The components of ZP were analyzed using GC-MS, and distilled extracts of ZP (deZP) were prepared. **Results:** In vitro, deZP decreased inflammatory cytokines. However, in vivo, it also decreased paw thickness and IL-1β levels. The anti-inflammatory effects of deZP are believed to be mediated through the NLRP3 inflammasome pathway, as indicated by RNA sequencing and network pharmacology analyses. **Conclusions:** ZP has an anti-inflammatory effect and regulation of the NLRP3 inflammasome in vitro and in vivo. Further research, including clinical trials, is required to confirm the safety of deZP, determine the optimal dosing, and evaluate its long-term effects.

## 1. Introduction

Gouty arthritis is a prevalent and chronic form of inflammatory arthritis that primarily affects adults, resulting from the deposition of monosodium urate (MSU) crystals. According to recent studies, its prevalence ranges from <1–6.8%, and its incidence ranges from 0.58 to 2.89 per 1000 person-years [1]. However, drugs prescribed to patients in the current clinical environment have limited effectiveness. For instance, previous studies have shown that more than 80% of patients who take colchicine experience abdominal pain before experiencing complete clinical improvement. Moreover, the side effects of non-steroidal anti-inflammatory drugs (NSAIDs) are more pronounced in the elderly [2]. Importantly, these drugs only offer temporary pain relief and do not provide a radical cure for gouty arthritis.

Gouty arthritis is characterized by the deposition of uric acid crystals in joints and periarticular tissues. Recent research has identified a receptor, NOD-like receptor family, pyrin domain containing 3 (NLRP3), that responds to uric acid crystals and triggers an inflammatory signal. NLRP3 is a member of the NOD-like receptor family (NLR), which detects endogenous danger signals, including microbial invasion and uric acid crystals. When these signals are present, NLRP3 combines with apoptosis-related speckle-like proteins, such as adapter proteins, CARD (ASC), and pro-caspase-1, to form an inflammasome. Pro-caspase-1 is then cleaved into an active form of caspase-1, which in turn cleaves the pro-IL-1β precursor to generate active IL-1β, which is secreted into the extra-cellular environment [3,4,5]. While current anti-IL-1β therapies are expected to be effective against acute gouty arthritis attacks, they are associated with high costs, inconvenient treatment routes and regimens, and side effects. Therefore, it is crucial to identify inflammatory mechanisms underlying the pathogenesis of gouty arthritis and to develop more effective therapeutic agents [6,7,8].

*Zanthoxylum piperitum* Benn. (ZP) is a summergreen shrub that is commonly found in East Asia, such as Korea, China, and Japan. ZP includes a lot of spices and oil ingredients in root, stem, leaf, and fruit [9]. Hence, ZP was reported to have been used for various purposes, including as an anti-inflammatory [10], antibacterial [11], and antioxidant agent [12], as well as oriental medicine used to treat disease such as osteoarthritis [12], obesity [13], and cancer [14]. Terpinen-4-ol, one of the active compounds of ZP, was previously studied for anti-inflammatory effects. It decreased pro-inflammatory cytokine production via nuclear factor kappa B (NF-kappa B) and NLRP3 inflammasome activation [15]. The mechanism against gouty arthritis, one of the inflammatory diseases, is not yet clear. We selected the distilled extracts method to use ZP as a pharmacopuncture.

Given that IL-1β is a major cytokine produced in response to MSU deposition and NLRP3 inflammasome activation is strongly linked to the development of gouty arthritis [12,16,17], targeting the NLRP3 inflammasome may be an effective therapeutic strategy for this condition. In this study, we aimed to investigate the effect of distilled extracts of ZP, which is known to have anti-inflammatory properties [9,18], on IL-1β production and inflammasome signal transduction in gouty arthritis.

## 2. Results

### 2.1. Anti-Inflammatory Effects of Destilled Extracts ZP (deZP) in Macrophages

To determine the potential cytotoxicity of deZP, we treated RAW264.7 cells with varying concentrations of deZP ranging from 0.32 to 5%. We observed that the highest concentration of 5% showed cytotoxicity in RAW264.7 cells. Consequently, we excluded this concentration from further investigations. Next, we examined whether deZP could exhibit anti-inflammatory effects in macrophages. We induced inflammation in RAW264.7 cells using lipopolysaccharide (LPS) and treated them with different concentrations of deZP. Our results showed that as the concentration of deZP increased, the levels of nitrite, IL-6, TNF-α, and MPO activity decreased in a concentration-dependent manner (Figure 1). This finding indicates that deZP has potential anti-inflammatory effects in macrophages.

### 2.2. Regulation of NLRP3 Inflammasome by deZP in Transcriptional Profiling and Gene Ontology

To investigate whether the anti-inflammatory efficacy of deZP is related to the NLRP3 inflammasome, we conducted RNA sequencing with bone marrow-derived macrophage (BMDM) cells. We induced the NLRP3 inflammasome by treating BMDM cells with MSU crystals after LPS priming. We then compared cells treated with or without deZP. Our results showed that genes that increased after MSU treatment in various categories were decreased after treatment with deZP (Figure 2A). A scatter plot showed that the genes were regulated by deZP in the inflammasome categories. It shows that increased NLRP3 gene expression by MSU regulates the decrease after treatment with deZP (Figure 2B). Unfortunately, IL-1β was not regulated by deZP. Additionally, gene ontology analysis revealed that deZP down-regulated genes involved in cytokine–cytokine receptor interaction and the NF-kappa B signaling pathway. These findings demonstrate that deZP can regulate NF-kappa B-related inflammatory responses, as well as the NLRP3 inflammasome (Figure 2C).

### 2.3. Association with the ZP’s Active Compound and Gouty Arthritis Through Network Construction

Because natural products such as ZP have various active compounds, we need to select one of ZP’s compounds to construct the network. A previous study reported that Terpinen-4-ol, as one of ZP’s compounds, has anti-inflammatory effects and relates with the NLRP3 inflammasome [15]. Hence, we construct the network with Terpinen-4-ol as the active compound. Moreover, 44 genes of the Terpinen-4-ol network with 104 nodes and 1434 edges (Figure 3A). To investigate the effects of the Terpinen-4-ol on gouty arthritis, the target genes of Terpinen-4-ol network and the ‘gout’ gene set were compared, resulting in the deduction of eight intersecting genes, including *NLRP3*, *PTGS2*, *CXCL8*, *IL6*, *TNF*, *IL1B*, *TLR4*, and *ALB* (Figure 3C). Although the matching percentage between Terpinen-4-ol-targeted genes and the gouty arthritis gene set was only 9.5% (Figure 3B), the common genes were highly linked with the 8 nodes and 28 edges around *NLRP3* (Figure 3D).

### 2.4. Prediction of the Potential Underlying Mechanism of ZP on Gouty Arthritis

Based on significant FDR values, the potential functional mechanisms of ZP on gouty arthritis were mainly associated with the ‘NOD-like receptor signaling pathway (hsa04621)’, ‘NF-kappa B signaling pathway (hsa04064)’, ‘Toll-like receptor signaling pathway (hsa04620)’, ‘TNF signaling pathway (hsa04668)’, and ‘Cytokine–cytokine receptor interaction (hsa04060)’ on the KEGG Pathways database (Figure 4A). Additionally, the GO Biological Process database showed the association with the ‘Regulation of acute inflammatory response (GO:0002673)’, ‘Response to lipopolysaccharide (GO:0032496)’, ‘Cellular response to lipopolysaccharide (GO:0071222)’, ‘Inflammatory response (GO:0006954)’, ‘Regulation of cytokine production involved in immune response (GO:0002718)’, ‘Regulation of inflammatory response (GO:0050727)’, and ‘Cytokine production (GO:0001816)’ with the underlying mechanisms of ZP (Figure 4C). Those predicted biological terms shared NLRP3, TLR4, IL6, IL1B, TNF, CXCL8, and PTGS2 in KEGG Pathways and GO Biological Process (Figure 4B,D).

### 2.5. Specific Mechanisms of ZP in NOD-like Receptor Signaling Pathway in Gouty Arthritis

With the purpose of finding a more specific mechanism of ZP, genes were compared with the ZP network and the ‘NOD-like receptor signaling pathway’, which is primarily listed in the prediction results of the KEGG Pathway above. On a map of the KEGG Pathway (hsa04621), the common genes were highlighted in a red box, showing that ZP-targeted genes particularly belong to ‘Pro-inflammatory cytokines’, ‘Chemokines’, ‘Toll-like receptor signaling pathway’, and ‘Pro-inflammatory effects’ (Figure 5).

### 2.6. Anti-Inflammation of deZP in Acute Gouty Arthritis Mouse Model

We investigated whether deZP could inhibit the NLRP3 inflammasome in a gouty arthritis mouse model. We orally administered deZP to mice and induced gouty arthritis by injecting MSU crystals into their footpads. We observed a statistically significant decrease in footpad thickness in mice treated with deZP from 6 h after MSU administration (Figure 6A). Moreover, IL-1β levels in the footpads of mice administered with deZP also showed a statistically significant decrease (Figure 6B). deZP decreased also caspase-1 and IL-1β protein levels in Western blot (Figure 6C). In real-time PCR, deZP also regulated IL-1β mRNA expression level as vehicle. However, NLRP3 and caspase-1 mRNA expression were not regulated by deZP (Figure 6D). These results demonstrate that deZP attenuated IL-1β mRNA expression and protein levels produced by the NLRP3 inflammasome in an acute gouty arthritis mouse model induced by MSU.

### 2.7. Ingredients of deZP

To investigate the ingredients contained in deZP, we performed electron ionization mass spectra. These ingredients include Linalool, Terpinen-4-ol, α-terpineol, Camphene, and Piperitone [19]. When we investigated the relative peak area on the total ion chromatogram for deZP, we found that Terpinen-4-ol, which appeared at 14.5 min on the chromatogram, was the most abundant component (Figure 7A). These ingredients were reported to have various functions. Ingredients of deZP are used antimicrobial, insecticide, and food additive [20,21,22,23]. Exceptionally, Delta 3-carene is useful to improve sleep and Piperitone is one of the main materials for synthetic Menthol and Thymol. Furthermore, we measured retention times and relative area abundances of peaks of each ingredient (Figure 7B) [21,24]. Terpinen-4-ol, the most frequently identified ingredient in deZP, was calculated for its quantification. Terpinen-4-ol was contained in about 487.36 μg/mL of deZP (Figure 7C).

## 3. Materials and Methods

### 3.1. Animal Care and Cell Culture

The Institutional Animal Care and Use Committee (IACUC) of Woosuk University of Korea approved the experimental procedures and animal care (Permission no. WS2020-05). C57BL/6 mice obtained Daehan bio (Seoul, Republic of Korea) were housed in a controlled animal facility with a 12:12-h light/dark cycle, a temperature of 23 ± 3 °C, and a relative humidity of 40–60%. We cultured BMDMs according to the previously described protocol [14]. Macrophages and RAW264.7 cells were cultured in Dulbecco’s modified Eagle medium supplemented with 10% (*v*/*v*) fetal bovine serum obtained Invitrogen (Carlsbad, CA, USA), 10,000 units/mL penicillin, and 10,000 μg/mL streptomycin.

### 3.2. Reagents

We obtained purified lipopolysaccharides (LPS) from Escherichia coli from List Bio-logical Laboratory (Campbell, CA, USA), which we dissolved in endotoxin-free water. We obtained MSU from Invivogen (Carlsbad, CA, USA), and we crystallized it in phosphate-buffered saline (PBS) for 5 min of sonication. We obtained the nitrite assay kit, protease inhibitor cocktail, and phosphatase inhibitor cocktail from Sigma-Aldrich (St. Louis, MO, USA). We obtained tissue protein extraction reagent (T-PER) from Thermo Scientific (Waltham, MA, USA). We purchased distilled extracts of ZP from AJ Pharmacopuncture (Seoul, Republic of Korea). deZP was mixed with 10 L of distilled water in the extractor and boiled at 105 °C for 2 h. ZP extracts were distilled with about 5 L of water at 107 °C. deZP was stored at 4 °C. For Western blot detection, the IL-1β antibody was obtained from R&D Systems (Minneapolis, MN, USA), the NLRP3 and Caspase-1 antibodies were sourced from AdipoGen (San Diego, CA, USA), and the β-actin antibody was procured from Santa Cruz Biotechnology (Dallas, TX, USA). The secondary antibodies were purchased from Cell Signaling Technology (Danvers, MA, USA).

### 3.3. Cell Viability

We measured cell viability using the Cell Proliferation Kit I (MTT) obtained from Roche (Basel, Switzerland). After treating RAW264.7 cells with deZP at concentrations of from 0.32% to 5% for 24 h, we measured cell viability using a microplate reader, the SpectraMax^®^ ABS, purchased from Molecular Devices (San Jose, CA, USA).

### 3.4. Myeloperoxidase (MPO) Activity Assay

We measured MPO activity in supernatants of raw264.7 cells treated with LPS with deZP 0.63%, 1.25%, and 2.5% using the MPO Colorimetric Activity Assay Kit obtained from Bio-Vision (Milpitas, CA, USA), following the manufacturer’s instructions.

### 3.5. Enzyme-Linked Immunosorbent Assays

We determined the levels of mouse IL-1β, IL-6, and TNF-α in supernatants of mice foot homogenate and raw264.7 cells treated with LPS 100 ng/mL with deZP at concentrations ranging from 0.63%, 1.25%, and 2.5% for 24 h. It was analyzed using the Duoset ELISA Kit obtained from R&D Systems (Minneapolis, MN, USA) and measured 450 nm observance with Molecular Devices SpectraMax^®^ ABS (San Jose, CA, USA).

### 3.6. RNA Sequencing

We activated BMDMs’ NLRP3 inflammasome using LPS priming and treated them with MSU with or without deZP. We extracted total RNAs from BMDMs using Trizol rea-gent obtained from Invitrogen (Carlsbad, CA, USA), following the manufacturer’s instructions. E-Biogen Inc. (Seoul, Republic of Korea) performed RNA quality and RNA sequencing assays.

### 3.7. Network Construction and Comparison with the Gene Set of Gouty Arthritis

The network of ZP was constructed by collecting the target genes of Terpine-4-ol, also known as 4-Carvomenthenol, since the most component of ZP in this study consisted of Terpine-4-ol. The target genes of Terpine-4-ol were obtained from the PubChem database “https://pubchem.ncbi.nlm.nih.gov/” (accessed on 7 July 2023). A total of 44 genes as chemical–gene co-occurrences in the literature created a network of ZP. Additionally, the gene set of gouty arthritis was collected from the MalaCards database “https://www.malacards.org/” (accessed on 7 July 2023) by searching a keyword ‘gout’ (Table 1). The gouty arthritis gene set was a total of 45 genes and compared with the ZP network. We constructed a new network with a total of 8 intersecting genes between ZP and gouty arthritis.

### 3.8. Functional Enrichment Analysis with KEGG Pathways and GO Biological Process Databases

The potential functional pathways of ZP on gouty arthritis were predicted in the Kyoto Encyclopedia of Genes and Genomes (KEGG) pathway 2020 human [25] and Gene Ontology (GO) Process databases using the Cytoscape String App (ver. 3.9.1). A total of 8 overlapped genes between ZP and gouty arthritis were categorized by each database, and the number of matching genes was counted according to the FDR values. Eventually, a diagram image of the ‘NOD-like receptor signaling pathway’ was obtained from KEGG Pathways. The matched genes were marked in a red box.

### 3.9. Gas Chromatography-Mass Spectrometry Conditions

We conducted chromatographic analysis using a DB-5MS capillary column, 30 m length × 0.25 mm inner diameter, 0.25 μm film thickness; J&W Scientific (Folsom, CA, USA) combined with an Agilent 6890 N gas chromatograph system (Palo Alto, CA, USA). Helium gas served as the carrier gas with a flow rate of 1 mL/min. In split mode (5:1), we set the injection port temperature to 280 °C and the injection volume to 1 μL. The oven temperature was programmed as follows: initial temperature at 50 °C for 5 min, increased to 200 °C at a rate of 8 °C/min and held for 5 min, then increased to 320 °C at a rate of 30 °C/min and held for 4 min. We carried out mass spectrometric determination using an Agilent 5975C mass selective detector equipped with a quadrupole analyzer and electron ionization source. The mass spectrometer was optimized with the following conditions: electron ionization energy, 70 eV; mass range in scan mode, *m*/*z* 50–550; ion source temperature, 230 °C; and transfer line temperature, 280 °C.

### 3.10. Acute Gouty Arthritis Model

The mice were randomly divided into four groups, with five mice per group. Colchicine or deZP was orally administered in 0.1 mL of sterilized water. One hour after administration, MSU crystals were injected into the footpad of the mice. After 12 h, foot tissue was homogenized in T-PER, and the supernatants were collected for ELISA. Moreover, foot tissue was also homogenized in Trizol reagent for mRNA extraction. cDNA synthesis using the iScript cDNA synthesis kit obtained Bio-Rad (Hercules, CA, USA) according to the manufacturer. cDNA was synthesized by adjusting mRNA to a concentration of 1000 ng/μL. After synthesis, the purity and concentration of the cDNA were re-evaluated. The synthesized cDNA was then used to perform real-time PCR with the TB Green^®^ Fast qPCR Mix. The primer sequences are as follows: IL-1b forward: 5′-GCAACTGTTCCTGAACTCAACT-3′; IL-1b reverse: 5′-ATCTTTTGGGGTCCGTCAACT-3′, NLRP3 forward: 5′-ATTACCCGCCCGAGAAAGG-3′; NLPR3 reverse: 5′-TCGCAGCAAAGATCCACACAG-3′, caspase-1 forward: 5′-ACAAGGCACGGGACCTATG-3′; caspase-1 reverse: 5′-TCCCAGTCAGTCCTGGAAATG-3′. The primers were obtained from Bioneer (Daejeon, Republic of Korea).

### 3.11. Western Blot

Footpad tissues were homogenized in 500 μL of T-PER buffer using a Bead Ruptor 12 purchased Omni Inc. (Kennesaw, GA, USA). The homogenate was centrifuged at 20,000× *g* for 10 min at 4 °C, and the clear upper phase was carefully transferred to a new tube. Protein concentration was determined using a BCA assay kit according to the manufacturer’s instructions. Protein samples were denatured by boiling at 95 °C for 5 min and then loaded onto SDS-PAGE gels. Electrophoresis was performed at 80 V for 30 min, followed by 120 V for 1 h. Proteins were subsequently transferred onto nitrocellulose membranes.

To block nonspecific binding, the membranes were incubated with 5% BSA for 1 h. After overnight incubation with primary antibodies, membranes were treated with secondary antibodies conjugated to horseradish peroxidase for 1 h. The chemiluminescent reaction was initiated using ECL solution, and protein detection was conducted using the Amersham ImageQuant 800 system (Cytiva, Amersham, UK).

### 3.12. Statistics

Data were presented as the mean ± SD. Statistical analysis was conducted using GraphPad Prism 5 software (GraphPad). Data were analyzed by one-way ANOVA followed by Tukey’s test. Statistical significance was set at *p* < 0.05.

## 4. Discussion

In this study, we demonstrated that deZP for pharmacopuncture reduced IL-1β levels and suggested that this effect may be mediated by the NLRP3 inflammasome signal pathway in a gouty arthritis mouse model. ZP is a summer green shrub from the Ructaceae family and is widely used in East Asia as a traditional spice and medicine due to its various active oil components [26]. Although ZP has been reported to have diverse therapeutic effects, including detoxifying, reducing hypertension, stroke, antibacterial and antioxidant, tyrosinase, and osteosarcoma proliferation control [27,28], its anti-inflammatory effect and therapeutic ability as a target for NLRP3 inflammasome have not been studied. Thus, we aimed to investigate the anti-inflammatory effect of deZP, a drug used in pharmacopuncture, on NLRP3 inflammasome-derived gouty arthritis via its effect on IL-1β regulation.

Firstly, we used network pharmacology to verify the association with 4-Terpinen-ol as the active compound of ZP-targeted genes and the gouty arthritis gene set, which could describe the complexity between compounds, target genes, and diseases, indicating the potential mechanism of action of multitarget medicinal herbs [29]. The target genes of Terpine-4-ol, a main compound of ZP, were overlapped with the gouty arthritis-related genes such as *NLRP3*, *PTGS2*, *CXCL8*, *IL6*, *TNF*, *IL1B*, *TLR4*, and *ALB*. Those genes constructed the network predicted to be related with the ‘NOD-like receptor signaling pathway’, ‘NF-kappa B signaling pathway’, ‘Toll-like receptor signaling pathway’, and others. Specifically, ‘Pro-inflammatory cytokines’, ‘Chemokines’, ‘Toll-like receptor signaling pathway’, and ‘Pro-inflammatory effects’ were predicted to associate with the potential target mechanisms of ZP.

Our study found that deZP administration reduced paw thickness, a symptom of acute gouty edema, and decreased IL-1β mRNA and protein levels in an MSU-induced gouty arthritis animal model. Previous studies have reported that ZP or 4-Terpinen-ol inhibits NF-κB and NLRP3 [15,30]. RNA sequencing analysis demonstrated a reduction in NLRP3 expression mediated by deZP. However, this result was not reproducible in the gouty model, where IL-1β mRNA expression was decreased. These findings suggest that deZP has the potential to regulate the mRNA expression of both NLRP3 and IL-1β. In gouty arthritis, MSU activates the NLRP3 inflammasome through the priming signal of NF-κB-dependent transcription of NLRP3 and pro-IL-1β, which triggers a downstream signal upon activation of the TLR 4 receptor [31,32]. This signal promotes the expression of inflammasome components such as NLRP3, pro-caspase-1, and pro-IL-1β. The second signal involved in the assembly and activation of the NLRP3 complex is induced by MSU. The ASC aggregate recruits CARD, which interacts with the CARD domain of pro-caspase-1 to promote caspase-1 activation and ASC specks [32,33]. Clustered pro-caspase-1 mediates self-cleavage and activation in the form of activated caspase-1, which cleaves precursors of IL-1β to generate activated forms of IL-1β. Once NLRP3 activation is initiated, the formation of ASC spec is considered an upstream indication of NLRP3 activation [34]. Although we did not directly investigate the expression level of direct biomarker proteins among NLRP3 pathways, such as NF-κB, ASC, and PYD, it was observed that deZP lowered IL-1β levels by regulating one of these pathways. Thus, additional studies are necessary to confirm the biomarker of NLRP3 based on single-component analysis of ZP.

In vitro experiments also demonstrated that deZP reduced inflammatory cytokines and factors in LPS-induced inflammation. LPS is one of the representative PAMPs that acts as a priming signal in NLRP3 inflammasome activation. Although LPS treatment alone does not activate the NLRP3 inflammasome, it was also confirmed that ZP has anti-inflammatory effects on other pathways, as reported in previous studies [9,12,26]. Thus, ZP is considered to have various anti-inflammatory effects that can be applied to the treatment of gouty arthritis.

In the TIC plot for deZP of Figure 7A, other minor peaks were observed around 10, 14, 17 and 20 min. However, these peaks could not be identified, since the mass spectra for these peaks could not be properly interpreted due to their low intensities and lack of match with the library database.

Numerous studies have aimed to find natural remedies that can treat gouty arthritis by regulating IL-1β through targeting the NLRP3 inflammasome [8]. This study highlights the potential of the ZP drug as an effective treatment that can be utilized in clinical practice. Moreover, further research on the safety and effectiveness of ZP in clinical studies, based on comprehensive mechanism research and preclinical studies, is necessary to establish it as an exceptional drug for gouty arthritis treatment.

## Figures and Tables

**Figure 1 pharmaceuticals-18-00029-f001:**
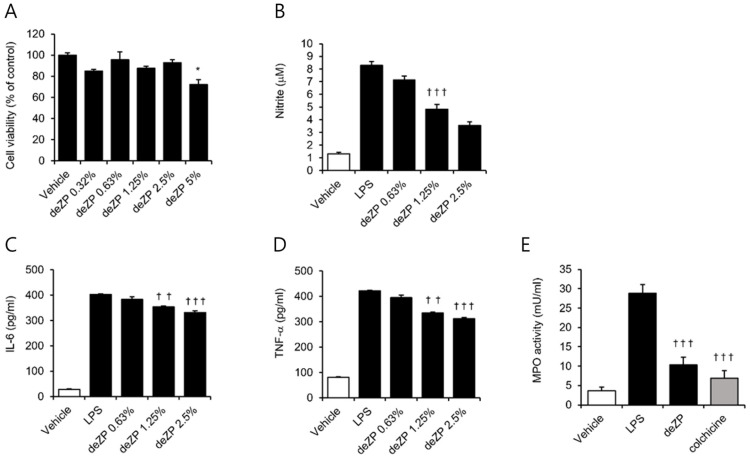
Anti-inflammatory effects of deZP. (**A**) RAW264.7 cells were treated with deZP at concentrations ranging from 0.32% to 5% for 24 h. (**B**–**D**) RAW264.7 cells were treated with deZP at concentrations of 0.63%, 1.25%, and 2.5% for 1 h, followed by stimulation with LPS (100 ng/mL) with deZP for 24 h. (**C**,**D**) Cell supernatants were collected for analysis of IL-6 and TNF-α using ELISA. (**E**) RAW264.7 cells were stimulated with LPS (100 ng/mL) and treated with deZP or colchicine (10 μM). The data are presented as mean ± SD (*n* = 3). * *p* < 0.05 vs. vehicle. †† *p* < 0.01, ††† *p* < 0.001 vs. LPS.

**Figure 2 pharmaceuticals-18-00029-f002:**
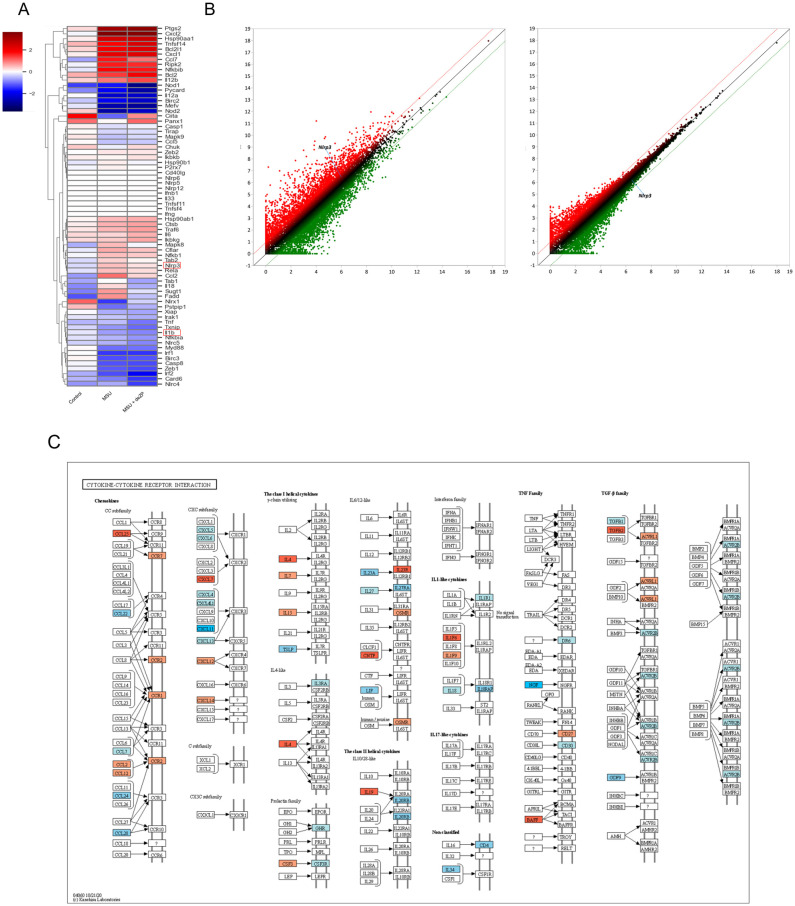
Analysis of significantly differentially expressed genes (DEGs) and gene ontology (GO). (**A**,**B**) Heatmap showing genes with differential expression and exhibiting a scatter plot of DEG. They show inflammasome categories. (**A**) red boxes are presented as target genes. (**B**) red spots mean up regulated genes, and green are down regulated genes. (**C**) Gene ontology shows regulated genes by deZP in specific signaling (cytokine–cytokine receptor interactions and NF-kappa B signaling pathway). Red boxes are up-regulated genes, and blue are down-regulated genes.

**Figure 3 pharmaceuticals-18-00029-f003:**
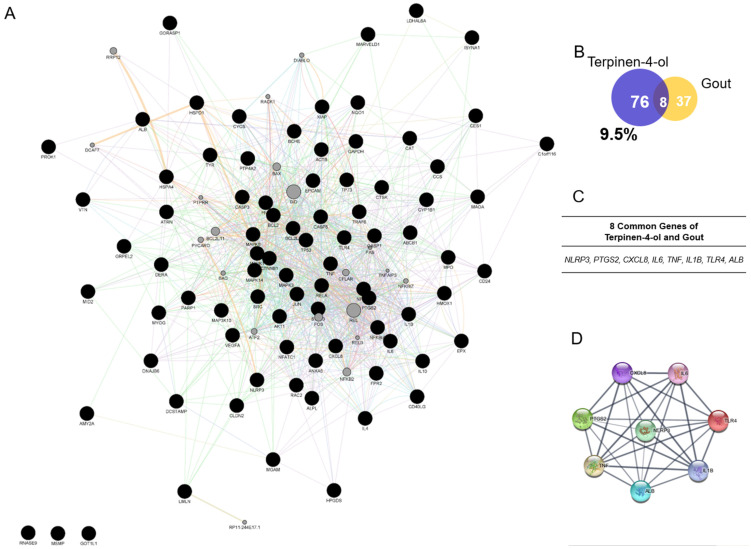
Network of Terpinen-4-ol as ZP’s active compound. (**A**) Whole network Terpine-4-ol as an active compound of ZP with 104 nodes and 1434 edges. (**B**) Venn diagram of intersection target genes between the Terpinen-4-ol network and ‘gout’ gene set. (**C**) Eight common genes of Terpinen-4-ol and gout. (**D**) Network of common genes of Terpinen-4-ol and gout with 8 nodes and 28 edges.

**Figure 4 pharmaceuticals-18-00029-f004:**
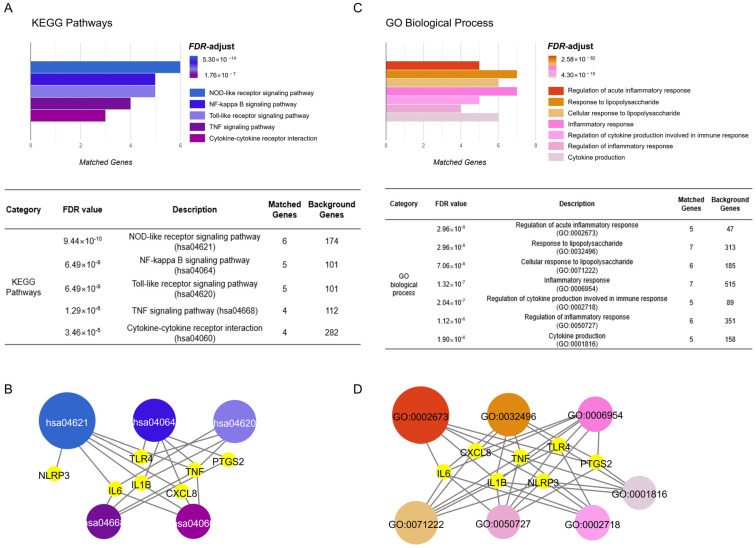
Functional enrichment analysis of the ZP network and ‘gout’ gene set. (**A**) Biological terms of the intersecting genes of ZP and gout using a KEGG Pathways database in high FDR value order. (**B**) Significantly enriched pathway terms of ZP in gout on a KEGG Pathways database. (**C**) Biological terms of the intersecting genes of ZP and gout using a GO Biological Process database in high FDR value order. (**D**) Significantly enriched pathway terms of ZP in gout on a GO Biological Process database.

**Figure 5 pharmaceuticals-18-00029-f005:**
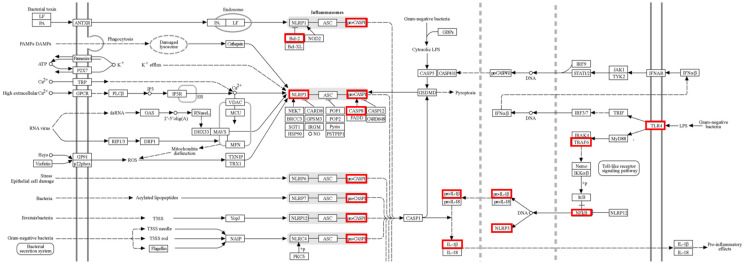
A diagram of the NOD-like receptor signaling pathway in the KEGG Pathway database (hsa04621). All matched genes are marked in a red box.

**Figure 6 pharmaceuticals-18-00029-f006:**
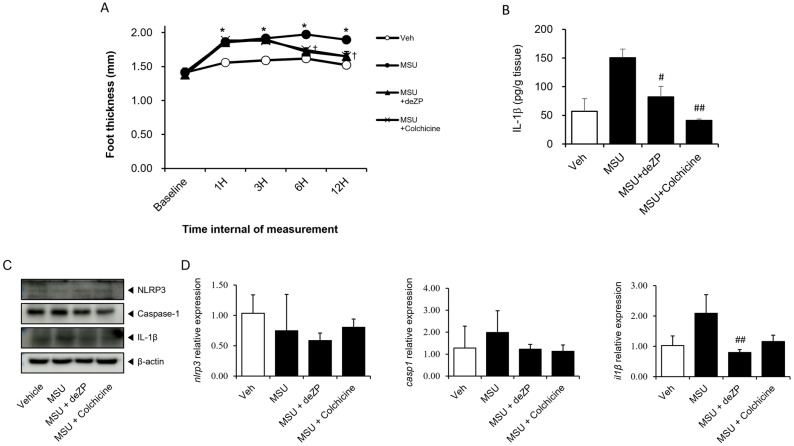
Treatment with deZP attenuates in vivo inflammatory symptoms. (**A**) Time course of footpad thickness after oral administration of deZP and injection of MSU crystals. (**B**) IL-1β levels in footpad tissue measured by ELISA. (**C**) Footpad tissues were analyzed by Western blot for NLRP3, caspase-1, and IL-1β. (**D**) Real-time PCR shows that mRNA expression levels of NLRP3, caspase-1, and IL-1β in footpad tissue. Data are presented as mean ± SD (*n* = 5 mice per group). * *p* < 0.05 vs. vehicle. † *p* < 0.05 vs. MSU. # *p* < 0.05, ## *p* < 0.01 vs. MSU.

**Figure 7 pharmaceuticals-18-00029-f007:**
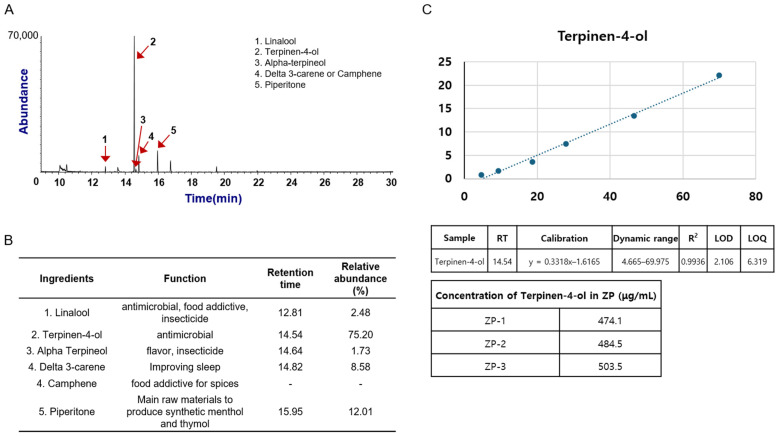
Identification of deZP and their functions. (**A**) Total ion chromatogram of deZP with labeled peaks: 1. Linalool; 2. Terpinen-4-ol; 3. Alpha-terpineol; 4. Delta 3-carene or Camphene; 5. Piperitone. (**B**) Function of the identified ingredients of deZP. (**C**) terpinene-4-ol, the main compound of ZP, was measured for concentrations.

**Table 1 pharmaceuticals-18-00029-t001:** List of the related target genes of ‘gout’ from the MalaCards database.

Description	Symbol	Relevance Score
Aspartyl-TRNA synthetase 2, mitochondrial	*DARS2*	400
Hypoxanthine phosphoribosyltransferase 1	*HPRT1*	61
Uromodulin	*UMOD*	41.1
Solute carrier family 22 member 12	*SLC22A12*	36.5
Xanthine dehydrogenase	*XDH*	34.63
Phosphoribosyl pyrophosphate synthetase 1	*PRPS1*	32.88
ATP binding cassette subfamily g member 2 (junior blood group)	*ABCG2*	26.2
Adenine phosphoribosyltransferase	*APRT*	25.02
MEFV innate immunity regulator, pyrin	*MEFV*	23.79
Interleukin 6	*IL6*	22.83
Interleukin 18	*IL18*	22.82
C-X-C Motif chemokine ligand 8	*CXCL8*	22.3
Major histocompatibility complex, class I, B	*HLA-B*	22.13
Solute carrier family 22 member 8	*SLC22A8*	22.05
Albumin	*ALB*	21.92
Interleukin 1 beta	*IL1B*	21.69
Purinergic receptor P2X 7	*P2RX7*	21.66
Solute carrier family 22 member 7	*SLC22A7*	20.29
Cytochrome P450 family 2 subfamily c member 8	*CYP2C8*	19.07
Cytochrome P450 family 2 subfamily E member 1	*CYP2E1*	18.79
Interleukin 1 receptor accessory protein like 2	*IL1RAPL2*	18.58
Polycystin 2, transient receptor potential cation channel	*PKD2*	18.41
Solute carrier family 2 member 9	*SLC2A9*	17.69
Purine nucleoside phosphorylase	*PNP*	16.83
NLR pyrin domain containing 3	*NLRP3*	16.64
Solute carrier family 17 member 1	*SLC17A1*	16.24
Adenylosuccinate lyase	*ADSL*	16.11
Solute carrier family 22 member 11	*SLC22A11*	16
Lactate dehydrogenase D	*LDHD*	15.72
Solute carrier family 17 member 3	*SLC17A3*	15.5
Tumor necrosis factor	*TNF*	15.3
PDZ domain containing 1	*PDZK1*	14.97
Solute carrier family 16 member 9	*SLC16A9*	14.87
Insulin	*INS*	14.78
Aldehyde dehydrogenase 2 family member	*ALDH2*	14.78
Solute carrier family 22 member 6	*SLC22A6*	14.55
S100 Calcium binding protein A9	*S100A9*	14.55
C-Reactive protein	*CRP*	14.31
Interleukin 1 receptor type 1	*IL1R1*	14.22
Alpha kinase 1	*ALPK1*	14.2
Capping protein regulator and myosin 1 linker 1	*CARMIL1*	14.18
Toll-like receptor 4	*TLR4*	14.17
Solute carrier family 22 member 13	*SLC22A13*	14.12
Prostaglandin-endoperoxide synthase 2	*PTGS2*	13.95
PYD and CARD domain containing	*PYCARD*	13.92

## Data Availability

The data presented in this study are available upon written and reasonable request directed at the corresponding authors.

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
