# Peer review of "Zanthoxylum piperitum Benn. Attenuates Monosodium Urate-Induced Gouty Arthritis: A Network Pharmacology Investigation of Its Anti-Inflammatory Mechanisms"

_pharmaceuticals, 2024, doi:10.3390/ph18010029_

Round 1
Reviewer 1 Report
Comments and Suggestions for Authors
I have reviewed the manuscript “Zanthoxylum piperitum Attenuates MSU-Induced Gouty Arthritis: A Network Pharmacology Investigation of Its Anti-Inflammatory Mechanisms.” The authors demonstrated the anti-inflammatory effect of the ZP extract on gout in mice. By the RNA-seq and the network pharmacological approaches, the authors demonstrated that the ZP extract expresses its anti-inflammatory effect by regulating the NLRP3 inflammasome. Finally, the authors demonstrated that the oral administration of ZP extract prevents monosodium urate-induced inflammation in mice. Although it was known that ZP extract and terpine-4-ol, its major ingredient, have an anti-inflammatory effect through NLRP3 inflammasome, the findings that the ZP extract can be a potential therapeutics for gout appear to be interesting and has enough impact on further research and the research field. However, I found several concerns about the display of the results and the conclusions the author argued in the manuscript. The critical issue is that the data displayed in Fig. 2, 3, and 5 were completely unreadable, hindering the understanding and evaluation of the manuscript. The authors need to provide a readable quality of the figures. My other specific comments are as follows.
Specific comments
1. The authors should adequately introduce the previous studies reporting that terpinen-4-ol exhibited an anti-inflammatory effect through NLRP3 inflammasome.
2. For Fig. 1 and 2, the authors should adequately refer to individual data instead of referring to all the data displayed in the figure.
3. It appears important to show the deZP treatment decreases IL-1b expression in the BMDM under the same condition as the cell subjected to RNA-Seq analyses.
4. The explanation and figure legend in Fig. 2(c) appears to be lacking.
5. The authors chose terpine-4-ol as an active compound for the network analyses. However, the authors are arguing and displaying the result from the network of terpine-4-ol as a network of ZP. This leads to misreading and is overstating. Please consider rephrasing them.
6. The authors concluded in Fig. 6 that “These results demonstrate that deZP attenuated IL-1β levels produced by NLRP3 inflammasome in an acute gouty arthritis mouse model induced by MSU (Fig. 6).” Do the results of the foot pad thickness and IL-1b amount indicate the contribution of NLRP3 inflammasome enough?
7. A detailed explanation for the data analyses of RNA-Seq is needed.
8. Please spell out all the abbreviations when they appear the first time in the main text except for the abstract.
9. Line 178, “exceptionally” should be “Exceptionally”
10. Line 181, “ug/ml” should be “µg/ml”
11. Line 201 to 202, please follow the journal’s citation style.
Author Response
Thank you very much for taking the time to review this manuscript.
We have addressed each of your comments as outlined below, and detailed explanations of the revisions are provided:
- The authors should adequately introduce the previous studies reporting that terpinen-4-ol exhibited an anti-inflammatory effect through NLRP3 inflammasome.
> We sincerely appreciate the reviewer’s insightful comments.
Through a literature search using the keywords “terpine-4-ol” and “NLRP3,” we identified previous studies that explored the active component of ZP, terpine-4-ol, in relation to the NLRP3 inflammasome. These findings further support the relevance of terpine-4-ol to the NLRP3 inflammasome mechanism. Based on this evidence, our choice of terpine-4-ol strengthens the premise of ZP’s efficacy in a gout model induced by MSU. Additionally, it highlights the potential of ZP as an anti-inflammatory therapeutic agent.
- For Fig. 1 and 2, the authors should adequately refer to individual data instead of referring to all the data displayed in the figure.
> We sincerely appreciate the reviewer’s thoughtful comments.
The cell viability data presented in Figure 1 were derived from in vitro experiments conducted to determine non-toxic concentrations for subsequent analyses. Furthermore, to confirm the reproducibility of the anti-inflammatory effects previously reported, we evaluated various markers derived from different mechanisms, including nitrite, IL-6, TNF-α, and MPO activity, in macrophages. These results allowed us to verify the potential of deZP as a therapeutic agent.
We agree that Figure 2 attempted to present too much information, which may have reduced focus and resolution. The heatmap and scatter plot alone sufficiently illustrate the reduction in NLRP3 inflammasome-related genes elevated by MSU under the influence of deZP. This constructive critique has guided us in improving the clarity and quality of our article, and we once again express our gratitude for the insightful comments.
- It appears important to show the deZP treatment decreases IL-1b expression in the BMDM under the same condition as the cell subjected to RNA-Seq analyses.
> We agree with the reviewer’s insightful suggestion regarding the need to demonstrate the reduction of IL-1β by deZP. Upon improving the resolution and re-evaluating the data, we noticed that IL-1β was not initially visible and therefore re-organized the data. While the reduction in IL-1β did not occur, we observed a decrease in the expression of NLRP3-related genes.
- The explanation and figure legend in Fig. 2(c) appears to be lacking.
> We sincerely thank the reviewer for their kind and constructive comments.
Upon reviewing the manuscript, we identified that the legend for Fig. 2(C) was inadvertently omitted. Fig. 2(C) presents data from GO analysis of differentially expressed genes after inducing inflammasome activation in macrophages with MSU, followed by treatment with deZP and RNA-seq analysis. We have added the missing explanation to the legend, clarifying that red indicates increased gene expression due to deZP, while blue represents decreased gene expression. This improvement, prompted by the reviewer’s valuable feedback, has helped us enhance the clarity and quality of the article, for which we are deeply grateful.
- The authors chose terpine-4-ol as an active compound for the network analyses. However, the authors are arguing and displaying the result from the network of terpine-4-ol as a network of ZP. This leads to misreading and is overstating. Please consider rephrasing them.
> Answer: We do agree with the reviewer. Because terpinen-4-ol is found to be an active component among the various compounds of ZP, we conducted a network analysis with the target genes of terpinen-4-ol. According to the reviewer’s comment, we have revised the term ‘network of ZP’ to the ‘network of terpinen-4-ol’ to avoid confusion.
- The authors concluded in Fig. 6 that “These results demonstrate that deZP attenuated IL-1β levels produced by NLRP3 inflammasome in an acute gouty arthritis mouse model induced by MSU (Fig. 6).” Do the results of the foot pad thickness and IL-1b amount indicate the contribution of NLRP3 inflammasome enough?
> We agree with the reviewer’s insightful comment.
One of the co-authors previously conducted experiments and reported findings on gout using a similar methodology. However, the observed reduction in IL-1β levels and paw thickness in that study was insufficient to conclusively demonstrate the contribution of the NLRP3 inflammasome. To address this, we conducted additional experiments using stored samples. Through real-time PCR analysis, we confirmed that IL-1β expression levels, which were elevated by MSU, were significantly reduced by deZP treatment. Western blot analysis further revealed that IL-1β expression was decreased by deZP, as was Caspase-1 expression. However, NLRP3 expression remained unchanged. These results demonstrate that deZP attenuates IL-1β and Caspase-1 in the NLRP3 inflammasome signal pathway. Once again, we sincerely appreciate your insightful suggestion.
- A detailed explanation for the data analyses of RNA-Seq is needed.
> We appreciate the reviewer’s valuable comments and agree with the feedback provided.
We identified issues with the resolution of the figure and the lack of sufficient explanation, which may have made data interpretation challenging. To address this, we refined the RNA-seq data by filtering out unnecessary information and retaining only the core data relevant to our findings. The resolution of the figure was also enhanced. Additionally, we highlighted key genes of interest, such as NLRP3 and IL-1β, using red boxes to guide the reader's focus. Detailed explanations regarding these changes have been added to both the main text and the figure legend. Once again, we express our gratitude to the reviewer for their insightful suggestions, which have greatly contributed to improving the quality of our article.
- Please spell out all the abbreviations when they appear the first time in the main text except for the abstract.
> We sincerely thank the reviewer for their kind and constructive comments.
To ensure clarity and accuracy, we conducted a thorough review of the manuscript to check for any abbreviations or typographical errors. Necessary corrections were made to enhance the overall readability and precision of the text.
- Line 178, “exceptionally” should be “Exceptionally”
> We sincerely thank the reviewer for their kind and constructive comments.
To ensure clarity and accuracy, we conducted a thorough review of the manuscript to check for any abbreviations or typographical errors. Necessary corrections were made to enhance the overall readability and precision of the text.
- Line 181, “ug/ml” should be “µg/ml”
> We sincerely thank the reviewer for their kind and constructive comments.
To ensure clarity and accuracy, we conducted a thorough review of the manuscript to check for any abbreviations or typographical errors. Necessary corrections were made to enhance the overall readability and precision of the text.
- Line 201 to 202, please follow the journal’s citation style.
> We sincerely thank the reviewer for their thoughtful comments.
We agree that some of the emphasis used in the manuscript may have been unnecessary. Accordingly, we have revised the text to align with the formatting guidelines and style preferences of the journal.
In addition to addressing these points, we have also made some minor edits to improve the overall clarity and consistency of the manuscript.
We appreciate your time and effort in reviewing our revised submission and look forward to your feedback.
Thank you again for your guidance and support.
Reviewer 2 Report
Comments and Suggestions for Authors
The authors explores the anti-inflammatory effects of Zanthoxylum piperitum (ZP) in a mouse model of gouty arthritis, highlighting its impact on the NLRP3 inflammasome pathway. The findings indicate that ZP reduces inflammation markers and IL-1β levels, suggesting its potential as a safer alternative treatment for gouty arthritis. The study is of immense importance and should be accepted after addressing a few comments outlined here.
1. The authors need to provide the GO cellular component and GO Molecular function associated with intersecting genes of ZP and gout. They can refer the article of Mohanty et al. (2024).
Mohanty D, Padhee S, Sahoo C, Jena S, Sahoo A, Panda PC, Nayak S, Ray A. Integrating network pharmacology and experimental verification to decipher the multitarget pharmacological mechanism of Cinnamomum zeylanicum essential oil in treating inflammation. Heliyon. 2024 Jan 30;10(2).
2. The authors need to provide peak area along with the retention time of the labelled peaks obtained using GC-MS.
Author Response
Thank you very much for taking the time to review this manuscript.
We have addressed each of your comments as outlined below, and detailed explanations of the revisions are provided:
- The authors need to provide the GO cellular component and GO Molecular function associated with intersecting genes of ZP and gout. They can refer the article of Mohanty et al. (2024).
Mohanty D, Padhee S, Sahoo C, Jena S, Sahoo A, Panda PC, Nayak S, Ray A. Integrating network pharmacology and experimental verification to decipher the multitarget pharmacological mechanism of Cinnamomum zeylanicum essential oil in treating inflammation. Heliyon. 2024 Jan 30;10(2).
> We do agree with the reviewer. Actually, we already analyzed the target mechanisms on the GO cellular component and GO Molecular function databases. However, GO cellular component database did not show the result. And only ‘Cytokine activity’ and ‘Cytokine receptor binding’ description was found in the GO Molecular function database. Those results are supposed due to small number of intersecting genes (only 8) between ZP and gout. Nevertheless, we will add the figures regarding on the GO cellular component and GO Molecular function databases, if reviewer thinks we should present these results. We will wait reviewer’s response. Again, we appreciate the reviewer’s accurate point.
- The authors need to provide peak area along with the retention time of the labelled peaks obtained using GC-MS.
> Thank you for your comments and suggestions. As the reviewer suggested, we provided the information for retention times and relative area abundances of peaks for bioactive substances in Figure 7 (b) in the revised version of the manuscript
In addition to addressing these points, we have also made some minor edits to improve the overall clarity and consistency of the manuscript.
We appreciate your time and effort in reviewing our revised submission and look forward to your feedback.
Thank you again for your guidance and support.
Reviewer 3 Report
Comments and Suggestions for Authors
In this study, the authors investigated the potential anti-inflammatory effects of ZP against NSU-induced gout using RNA seq, in vitro and in vivo studies as well as network pharmacology. The experimental studies are well designed. I have the following comments.
· why did the authors used RAW cells not THP of human origin?
· The authors often used genes increased (as in line 96), do they mean increased gene expression or copy number?
· Did the authors measure IL-1 b in serum?
· Please indicate what is the relevance score in table ? And how the FDR was calculated?
· Did the authors confirm any of the RNA-seq data with Real-Time PCR ?
· Please check grammar and language of the manuscript
Comments on the Quality of English Language
Needs improvement
Author Response
Thank you very much for taking the time to review this manuscript.
We have addressed each of your comments as outlined below, and detailed explanations of the revisions are provided:
- why did the authors used RAW cells not THP of human origin?
> We appreciate the reviewer’s insightful comment.
In this study, we initially conducted experiments in animal models and subsequently utilized mouse-derived macrophages, as we believe these cells are more suitable for accurately elucidating the mechanism compared to human cells. However, we recognize the importance of confirming whether ZP functions through the same mechanism in human-derived cells for its development as a therapeutic agent. As such, we plan to perform follow-up studies using THP-1 cells to validate these findings.
- The authors often used genes increased (as in line 96), do they mean increased gene expression or copy number?
> We sincerely thank the reviewer for their valuable feedback.
We acknowledge that the explanation of gene expression regulation was insufficient, particularly for Fig. 2, which includes extensive data requiring clear interpretation. To address this, we removed unnecessary data and focused on presenting the changes in gene expression using heatmaps and scatter plots. Additionally, we provided detailed explanations on how deZP modulates key genes related to the NLRP3 inflammasome, which are central to the findings of this study.
- Did the authors measure IL-1 b in serum?
> We sincerely appreciate the reviewer’s insightful comments.
We conducted an IL-1β ELISA assay using serum samples; however, IL-1β was not detected in the serum. We believe this is likely due to the localized injection of MSU into the paw, which may not have triggered a systemic inflammatory response.
- Please indicate what is the relevance score in table ? And how the FDR was calculated?
> We appreciate the reviewer’s sincere comment.
The relevance score of ‘gout’-targeted genes in Table is derived from GeneCards database. According to the GeneCards Search Guide, the scoring mechanism is based on the ‘Theory Behind Relevance Scoring’, a formula to find matching documents and use the Boolean model. It is well established that the relevance score is important on the different resources associating the gene with the disease, showing the elite genes related to specific disease.
In addition, FDR value, which provides important statistical properties in network analysis, is automatically calculated from Functional enrichment analysis in Cytoscape software. We hope this revision could satisfy the reviewer. Please refer to the references below.
Practical Guide to Life Science Databases. Chapter, The GeneCards Suite. 2022, p27-56.
Clinical and Translational Science. Chapter 20, Epidemiologic and Population Genetic Studies. 2009, p289-299.
- Did the authors confirm any of the RNA-seq data with Real-Time PCR ?
> We sincerely thank the reviewer for their insightful comments.
To validate the RNA-seq data, we performed real-time PCR using mRNA extracted from the paw tissues of the experimental animals. cDNA was synthesized, and the expression levels of representative genes of the NLRP3 inflammasome, including NLRP3, Caspase-1, and IL-1β, were analyzed. While IL-1β expression was found to be significantly reduced in real-time PCR, the results did not fully align with the RNA-seq data. However, the observed reduction of NLRP3 gene expression in RNA-seq, coupled with the IL-1β reduction in real-time PCR, collectively supports the conclusion that IL-1β cytokine levels are indeed reduced.
- Please check grammar and language of the manuscript
> We sincerely thank the reviewer for their kind and constructive comments.
To ensure clarity and accuracy, we conducted a thorough review of the manuscript to check for any abbreviations or typographical errors. Necessary corrections were made to enhance the overall readability and precision of the text.
In addition to addressing these points, we have also made some minor edits to improve the overall clarity and consistency of the manuscript.
We appreciate your time and effort in reviewing our revised submission and look forward to your feedback.
Thank you again for your guidance and support.
Reviewer 4 Report
Comments and Suggestions for Authors
In the present study, the authors examined the anti-inflammatory mechanism of Zanthoxylum piperitum (ZP) through in vivo and in vitro experiments, as well as network pharmacology. The topic is of interest, and the conclusions are well-supported by the conducted studies. The paper is generally well-written and organized. However, some minor revisions are required before it is ready for publication:
- Line 42: Correct the typo "an-ti-inflammatory"; similarly, correct "ac-tivity" in line 81.
- Ensure that all acronyms are defined when first mentioned in the text.
- The clarity of the figures, particularly Figures 2 and 5, needs improvement. Please provide higher-quality versions. You may consider moving some of the less critical figures to the Supplementary Information (SI) and retaining only the most relevant ones in the main text.
- Section 2.7 contains numerous typos. Please proofread this section thoroughly.
- Line 187: Insert "of" before "ZP."
- Lines 201-202: Move the reference to the reference list at the end of the paper.
- Line 204: Correct "con-structed."
- There are additional typos throughout the manuscript. A thorough proofreading of the entire paper is needed.
Author Response
Thank you very much for taking the time to review this manuscript.
We have addressed each of your comments as outlined below, and detailed explanations of the revisions are provided:
- Line 42: Correct the typo "an-ti-inflammatory"; similarly, correct "ac-tivity" in line 81.
> We sincerely thank the reviewer for their kind and constructive comments.
To ensure clarity and accuracy, we conducted a thorough review of the manuscript to check for any abbreviations or typographical errors. Necessary corrections were made to enhance the overall readability and precision of the text.
- Ensure that all acronyms are defined when first mentioned in the text.
> We sincerely thank the reviewer for their kind and constructive comments.
To ensure clarity and accuracy, we conducted a thorough review of the manuscript to check for any abbreviations or typographical errors. Necessary corrections were made to enhance the overall readability and precision of the text.
- The clarity of the figures, particularly Figures 2 and 5, needs improvement. Please provide higher-quality versions. You may consider moving some of the less critical figures to the Supplementary Information (SI) and retaining only the most relevant ones in the main text.
> We appreciate the reviewer’s insightful comments and agree with the observations.
Figures 2 and 5 indeed contained excessive information, which led to resolution issues. To address this, we reorganized the data and removed unnecessary elements to streamline the presentation. Additionally, we enhanced the resolution of the images to ensure clarity and improve the overall readability of the figures.
Section 2.7 contains numerous typos. Please proofread this section thoroughly.
> We sincerely thank the reviewer for their kind and constructive comments.
To ensure clarity and accuracy, we conducted a thorough review of the manuscript to check for any abbreviations or typographical errors. Necessary corrections were made to enhance the overall readability and precision of the text.
- Line 187: Insert "of" before "ZP."
> We sincerely thank the reviewer for their kind and constructive comments.
To ensure clarity and accuracy, we conducted a thorough review of the manuscript to check for any abbreviations or typographical errors. Necessary corrections were made to enhance the overall readability and precision of the text.
- Lines 201-202: Move the reference to the reference list at the end of the paper.
> We sincerely thank the reviewer for their thoughtful comments.
We agree that some of the emphasis used in the manuscript may have been unnecessary. Accordingly, we have revised the text to align with the formatting guidelines and style preferences of the journal.
- Line 204: Correct "con-structed."
> We sincerely thank the reviewer for their kind and constructive comments.
To ensure clarity and accuracy, we conducted a thorough review of the manuscript to check for any abbreviations or typographical errors. Necessary corrections were made to enhance the overall readability and precision of the text.
- There are additional typos throughout the manuscript. A thorough proofreading of the entire paper is needed.
> We sincerely thank the reviewer for their kind and constructive comments.
To ensure clarity and accuracy, we conducted a thorough review of the manuscript to check for any abbreviations or typographical errors. Necessary corrections were made to enhance the overall readability and precision of the text.
In addition to addressing these points, we have also made some minor edits to improve the overall clarity and consistency of the manuscript.
We appreciate your time and effort in reviewing our revised submission and look forward to your feedback.
Thank you again for your guidance and support.
Round 2
Reviewer 1 Report
Comments and Suggestions for Authors
I have reviewed the revised version of the manuscript “Zanthoxylum piperitum Attenuates MSU-Induced Gouty Arthritis: A Network Pharmacology Investigation of Its Anti-Inflammatory Mechanisms.” The authors adequately addressed some of my concerns. However, there are still some concerns regarding the data display and conclusion.
Specific comments
1. The figures' quality was improved but still hard to read. In particular, Fig. 3a is totally invisible and thus meaningless. Also, it is unclear which plot indicates NLRP3 in the revised Fig. 2B.
2. The legend of Fig. 6C appears not to be correct. The data shows the result of western blots.
3. The western blot results displayed in Fig. 6C and D contradict the result of the RNA-seq study displayed in Fig. 2. As the author mentions, deZP treatment reduces mRNA expression of NLRP3 in the cell study but the protein expression is unchanged in the animal study. This might reflect the fact that deZP affects the mRNA expression of NLRP3 but not protein expression. The author should further discuss and rationalize how deZP attenuates NLRP3-related inflammation without affecting NLRP3 protein expression.
Author Response
Thank you very much for taking the time to review this manuscript.
We have addressed each of your comments as outlined below, and detailed explanations of the revisions are provided:
- The figures' quality was improved but still hard to read. In particular, Fig. 3a is totally invisible and thus meaningless. Also, it is unclear which plot indicates NLRP3 in the revised Fig. 2B.
> We fully agree with your insightful comment.
In attempting to present the full figures, we inadvertently made them difficult to view. To solve this, we have maximized the resolution of Fig. 2 and Fig. 3 using the original software and enlarged Fig. 2B and 3A for improved clarity.
- The legend of Fig. 6C appears not to be correct. The data shows the result of western blots.
> Thank you for your careful observation. We have identified and corrected the mislabeling in the legend for Fig. 6C and 6D to accurately reflect the corresponding data.
- The western blot results displayed in Fig. 6C and D contradict the result of the RNA-seq study displayed in Fig. 2. As the author mentions, deZP treatment reduces mRNA expression of NLRP3 in the cell study but the protein expression is unchanged in the animal study. This might reflect the fact that deZP affects the mRNA expression of NLRP3 but not protein expression. The author should further discuss and rationalize how deZP attenuates NLRP3-related inflammation without affecting NLRP3 protein expression.
> We sincerely thank you for your insightful comment.
We acknowledge that the RNA-seq data in Fig. 2 and the rtPCR data in Fig. 6D do not align perfectly. This discrepancy will require further validation through additional experiments. We suggest hypothesis that the traditional herbal material used in this study contains various compounds, which may undergo metabolic transformations in vivo. Consequently, additional studies are necessary to confirm whether Terpinen-4-ol or other compounds produce reproducible results in gout animal models, as seen in Fig. 6D.
We appreciate your time and effort in reviewing our revised submission and look forward to your feedback.
Thank you again for your guidance and support.
Round 3
Reviewer 1 Report
Comments and Suggestions for Authors
Regarding my previous comment #3, the author still needs to discuss in the manuscript possible reasons why NLRP3 protein and mRNA expression were unchanged in the mouse model. I understand that further experiments are needed, but without discussing this point, the author cannot argue that "These findings suggest that 228 deZP has the potential to regulate the mRNA expression of both NLRP3 and IL-1β".
Author Response
Dear Dr. Alexander George Panossian,
We thank you for the opportunity to revise our manuscript entitled "Zanthoxylum piperitum Attenuates MSU-Induced Gouty Arthritis: A Network Pharmacology Investigation of Its Anti-Inflammatory Mechanisms". We sincerely appreciate the reviewers' thoughtful and constructive feedback, which has significantly improved the clarity and quality of our work. Below, we provide a point-by-point response to the reviewer's comment and detail the corresponding revisions made to the manuscript.
Response to Reviewers’ Comments
Reviewer’s Comment #3:
“The author still needs to discuss in the manuscript possible reasons why NLRP3 protein and mRNA expression were unchanged in the mouse model. I understand that further experiments are needed, but without discussing this point, the author cannot argue that ‘These findings suggest that deZP has the potential to regulate the mRNA expression of both NLRP3 and IL-1β.”
Response:
We appreciate the reviewer’s valuable feedback regarding the need to discuss potential reasons why NLRP3 protein and mRNA expression were unchanged in the mouse model. Based on the reviewer’s suggestion, we have thoroughly revised the manuscript to address this issue in greater detail. Below, we provide an explanation for how deZP may attenuate NLRP3-mediated inflammation despite the observed lack of changes in NLRP3 mRNA and protein expression.
First, we hypothesize that deZP modulates NLRP3 activity primarily through post-transcriptional and post-translational mechanisms. While NLRP3 protein levels remained unchanged, it is well established that NLRP3 activation is largely regulated by post-translational modifications (PTMs), such as phosphorylation and ubiquitination, rather than by absolute changes in protein abundance. These PTMs are critical for the assembly, activation, and function of the NLRP3 inflammasome. We propose that deZP may interfere with these PTM processes, thereby attenuating NLRP3 activation without altering NLRP3 protein expression. Additionally, NLRP3 activation depends on the assembly of the inflammasome complex, which involves the recruitment of ASC, the aggregation of ASC specks, and interactions with CARD domains of pro-caspase-1. The formation of these complexes is essential for NLRP3 inflammasome function and IL-1β maturation. DeZP may regulate inflammation by inhibiting the assembly or stability of this inflammasome complex rather than by directly affecting NLRP3 protein expression.
Second, deZP may act on upstream pathways that regulate NLRP3 activation. For example, deZP could modulate NF-κB signaling, which provides the priming signal for NLRP3 and pro-IL-1β transcription. Although we did not directly measure biomarker proteins such as NF-κB, ASC, or PYD in this study, our data showed that deZP reduced IL-1β mRNA and protein levels, suggesting that deZP likely acts on one or more pathways associated with inflammasome priming or activation.
We have incorporated this discussion into the revised manuscript to provide a more comprehensive explanation of how deZP modulates NLRP3-related inflammation. We have also clarified that further studies are necessary to confirm the precise mechanisms by which deZP affects NLRP3 activation, particularly regarding PTMs and inflammasome assembly.
Lastly, we have adjusted the language in the manuscript to ensure that the conclusions are appropriately nuanced. Specifically, we revised the statement “These findings suggest that deZP has the potential to regulate the mRNA expression of both NLRP3 and IL-1β” to reflect that deZP may regulate NLRP3-related inflammation through mechanisms beyond changes in mRNA or protein expression.
These revisions are presented in the Discussion section (page 9, line 226-244) as follows:
“RNA sequencing analysis revealed that deZP treatment reduced NLRP3 mRNA expression. However, this reduction was not observed in the gouty arthritis model, where only IL-1β mRNA expression was decreased. While the mRNA and protein levels of NLRP3 remained unchanged in the gouty arthritis model, this does not rule out the possibility of functional modulation of the inflammasome. One plausible explanation is that deZP regulates post-translational modifications (PTMs) of NLRP3, which are critical for its activation. NLRP3 activation is primarily governed by PTMs, such as phosphorylation and ubiquitination. Therefore, deZP may attenuate NLRP3-mediated inflammation by interfering with these activation processes rather than altering the abundance of NLRP3 protein itself. Additionally, it is possible that deZP modulates upstream signaling required for inflammasome activation or specifically targets these PTMs to mitigate NLRP3-related inflammation. Consequently, the functional outcomes of deZP treatment could occur independently of changes in NLRP3 protein levels. NLRP3 activation also depends on the assembly of the inflammasome complex. This process involves the recruitment of ASC, which aggregates and recruits CARD domains that interact with pro-caspase-1, ultimately leading to its activation [31, 32]. Activated caspase-1 cleaves pro-IL-1β into its mature form, which mediates the inflammatory response. Importantly, the formation of ASC specks is considered an upstream indicator of NLRP3 inflammasome activation [33]. Thus, deZP may attenuate the assembly or expression of NLRP3 complex components without directly altering NLRP3 protein levels. Although we did not directly measure the expression of specific biomarker proteins in the NLRP3 pathway, such as NF-κB, ASC, or PYD, deZP was observed to reduce IL-1β levels, likely through the regulation of one or more of these pathways. These findings underscore the need for further studies to confirm the involvement of deZP in modulating NLRP3 through PTMs or by interfering with the formation of the NLRP3 inflammasome complex.”
We hope that this expanded discussion adequately addresses the reviewer’s concern.
Additionally, we have confirmed the checklist as follows:
(I) Relevance of References: We have reviewed all cited references to ensure they are relevant to the manuscript's content. No irrelevant references in the revised version.
(II) Highlighting Revisions: All revisions made to the manuscript, including changes to the Discussion sections, have been clearly highlighted in red to facilitate review by the editors and reviewers.
(III) Point-by-Point Response: This cover letter provides a detailed, point-by-point explanation of how we addressed the reviewers’ comments.
(IV) Reviewer-Recommended References: The reviewer did not suggest additional references. Therefore, no new references were added.
(V) Addressing Unresolved Comments: We have thoroughly addressed all reviewer comments. While we acknowledge that additional experimental validation (e.g., assessing microRNA involvement or NLRP3 PTMs) would strengthen our findings, such experiments are beyond the scope of this current study. We have clearly stated this limitation in the Discussion and proposed these experiments as future research directions.
Understanding the importance of prompt revisions for publication within the desired timeline, we have expedited our response. We believe that our manuscript, with the latest revisions and adherence to the author checklist, provides a comprehensive and clearer presentation of our research findings. We are confident that this manuscript will make a significant contribution to the field and aligns well with the high standards of Pharmaceuticals.
Thank you for considering our revised manuscript for publication. We are happy to provide any additional information or clarification if needed and look forward to the opportunity to contribute to the scientific community through your esteemed journal.
Sincerely,
Gabsik Yang, Ph.D.
Professor
College of Korean Medicine
Woosuk University
61 Seonneomeo 3 Gil
Jeonju, Jeonbuk
Republic of Korea, 54986
Tel: 82-063-290-9030
E-mail: yanggs@woosuk.ac.kr
